# chromMAGMA: regulatory element-centric interrogation of risk variants

Robbin Nameki[1,2,]*, Anamay Shetty[1,2,3,]*, Eileen Dareng[4], Jonathan Tyrer[4,5], Xianzhi Lin[1,2], The Ovarian Cancer Association Consortium, Paul Pharoah[4,5], Rosario I Corona[1,2], Siddhartha Kar[8,9,]‡, Kate Lawrenson[1,2,6,7,]‡

Candidate causal risk variants from genome-wide association studies reside almost exclusively in noncoding regions of the genome and innovative approaches are necessary to understand their biological function. Multi-marker analysis of genomic annotation (MAGMA) is a widely used program that nominates candidate risk genes by mapping single-nucleotide polymorphism summary statistics from genome-wide association studies to gene bodies. We augmented MAGMA to create chromatin-MAGMA (chromMAGMA), a method to nominate candidate risk genes based on the presence of risk variants within noncoding regulatory elements (REs). We applied chromMAGMA to a genetic susceptibility dataset for epithelial ovarian cancer (EOC), a rare gynecologic malignancy characterized by high mortality. This identified 155 unique candidate EOC risk genes across five EOC histotypes; 83% (105/127) of high-grade serous ovarian cancer risk genes had not previously been implicated in this EOC histotype. Risk genes nominated by chromMAGMA converged on mRNA splicing and transcriptional dysregulation pathways. chromMAGMA is a pipeline that nominates candidate risk genes through a gene regulation-focused approach and helps interpret the biological mechanism of noncoding risk variants for complex diseases.

## Introduction

Genome-wide association studies (GWASs) have identified thousands of germline single-nucleotide polymorphisms (SNPs) associated with myriad diseases and phenotypes [1]. Risk SNPs rarely exert their impact by altering the amino acid sequence of a protein-coding gene; it is now clear that a large proportion of risk SNPs modify the activity of noncoding regulatory elements such as transcriptional enhancers [2, 3, 4, 5]. Because regulatory elements often interact with target promoters over large genomic distances, identifying the gene targets of risk SNPs remains a major challenge in post-GWAS functional studies.

One commonly used tool to study genes and pathways associated with risk is the multi-marker analysis of genomic annotation, or MAGMA [6] software, which uses multiple regression to group individual-level or summary SNP association from GWASs to the level of genes while accounting for linkage disequilibrium (LD) between variants. Instead of testing millions of variants individually, MAGMA reduces the multiple testing burden by performing gene-level analyses and has emerged as a powerful approach for the discovery of candidate genes and pathways associated with risk of complex traits [7, 8, 9]. MAGMA captures SNPs positionally mapped to gene bodies; however, many studies have now shown that noncoding tissue-specific REs (such as transcriptional enhancers marked by H3K27ac) are enriched for risk SNPs, and risk REs often interact with genes hundreds of kilobases away [2, 3, 4, 5]. We therefore created a bioinformatic tool termed "chromatin-MAGMA," or chromMAGMA, a pipeline that augments MAGMA to infer the target gene of noncoding risk variants based on user-inputted disease-relevant REs and RE-to-gene maps. chromMAGMA nominates candidate risk genes informed by SNPs enriched in up or downstream noncoding regulatory elements rather than gene bodies.

Here we tested chromMAGMA in epithelial ovarian cancer (EOC), a deadly disease with ~22,240 new cases and 14,070 annual deaths in the United States [10]. EOC can be stratified into five main histologic subtypes (histotypes): high-grade serous (HGSOC), low-grade serous (LGSOC), endometrioid (EnOC), clear cell (CCOC), and mucinous ovarian cancer (MOC) [10, 11]. Each histotype is characterized by distinct molecular drivers, clinicopathologic features, and distinct germline genetic risk variants [12, 13]. Of the 39 known unique EOC susceptibility

[1]Women's Cancer Research Program at the Samuel Oschin Comprehensive Cancer Center, Cedars-Sinai Medical Center, Los Angeles, CA, USA [2]Division of Gynecologic Oncology, Department of Obstetrics and Gynecology, Cedars-Sinai Medical Center, Los Angeles, CA, USA [3]School of Clinical Medicine, University of Cambridge, Cambridge, UK [4]Centre for Cancer Genetic Epidemiology, Department of Public Health and Primary Care, University of Cambridge, Cambridge, UK [5]Centre for Cancer Genetic Epidemiology, Department of Oncology, University of Cambridge, Cambridge, UK [6]Cancer Prevention and Control Program, Samuel Oschin Comprehensive Cancer Center, Cedars-Sinai Medical Center, Los Angeles, CA, USA [7]Center for Bioinformatics and Functional Genomics, Cedars-Sinai Medical Center, Los Angeles, CA, USA [8]Medical Research Council Integrative Epidemiology Unit, University of Bristol, Bristol, UK [9]Population Health Sciences, Bristol Medical School, University of Bristol, Bristol, UK

Correspondence: kate.lawrenson@cshs.org
*Robbin Nameki and Anamay Shetty contributed equally to this work.
‡Siddhartha Kar and Kate Lawrenson jointly directed the study.

loci ($P$-value < $5 \times 10^{-8}$) identified through GWASs, nine are associated with risk of HGSOC, five with risk of LGSOC, four with risk of MOC, and one with risk of EnOC. 20 loci are associated with all invasive disease, or a combination of one or more histotypes (13, 14, 15, 16, 17, 18, 19, 20, 21, 22, 23, 24). These genome-wide significant risk loci represent a fraction of all narrow-sense heritability in EOC and it is predicted that additional SNPs also contribute to disease susceptibility (25, 26). Innovative approaches are needed to deconvolute additional true risk loci falling below genome-wide significance from false positives because of limited power, particularly for the rarer histotypes.

We applied chromMAGMA to EOC, inputting histotype-specific GWAS summary statistics, REs identified by H3K27ac chromatin immunoprecipitation-sequencing (ChIP-Seq) of Müllerian tissues, and RE-to-gene maps from the GeneHancer database (27). ChromMAGMA highlighted mRNA splicing and transcriptional dysregulation in EOC risk. In addition, active transcription factors (TFs) marked by super-enhancers (large stretches of active chromatin) were particularly enriched for EOC risk associations based on chromMAGMA analyses and are likely to represent the nexus of noncoding EOC risk and transcriptional dysregulation. Overall chromMAGMA offers a flexible, gene regulation-focused approach to nominate noncoding regulatory elements and target genes involved in risk of polygenic traits.

## Results

### chromMAGMA maps risk-associated, active regulatory elements to target genes

To survey risk SNPs in regulatory elements, rather than gene bodies, we built the chromMAGMA pipeline by modifying the pre-processing and processing steps of MAGMA. We tested the performance of chromMAGMA using GWAS summary statistics and epigenome data for EOC (Table S1), as follows: first, the genome is trimmed to only include regions annotated as high-confidence active REs from the GeneHancer database (28). This reduces the genome from three billion base pairs (bp) to ~400 million bp (Fig 1A). Because GeneHancer includes data from 46 tissue types, for an EOC-specific analysis, we then restricted the universe of Gene-Hancer REs to those regions marked by H3K27ac in normal and malignant Müllerian tissues and cell lines (27, 28, 29). This created a universe of ~200 million base pairs containing only regions of active chromatin identified in ovarian cancer-relevant tissues, and the likely target gene(s) for each RE. GWAS SNP identifiers (reference SNP cluster identifiers, rsIDs) from six histotype-specific GWAS summary statistics (CCOC, EnOC, HGSOC, LGSOC, MOC, and NMOC—a dataset consisting of all samples except for MOC) (Coetzee S, Dareng EO, Peng P, Rosenow W, Tyrer JP (2021) Integrative multi-omics analyses to identify the genetic and functional mechanisms underlying ovarian cancer risk regions. (Submitted for Publication) (Fig 1B) were then positionally mapped to the aforementioned RE dataset by applying the MAGMA annotation command (see the Materials and Methods section). The SNP rsID-to-RE annotation was then processed for gene-level analysis using MAGMA (see the Materials and Methods section) alongside EOC GWAS SNP summary statistics ($P$-values) and 1000 Genome European panel reference LD data (30). As multiple REs can regulate one gene (31), each gene was assigned the $P$-value of the most significant RE. We noted a positive correlation between the gene's $P$-value and the number of REs associated with each gene (Uncorrected Spearman's $\rho$ = 0.56, Fig S1) and this correlation was reduced after adjusting for multiple regulatory elements assigned to a gene (Corrected Spearman's

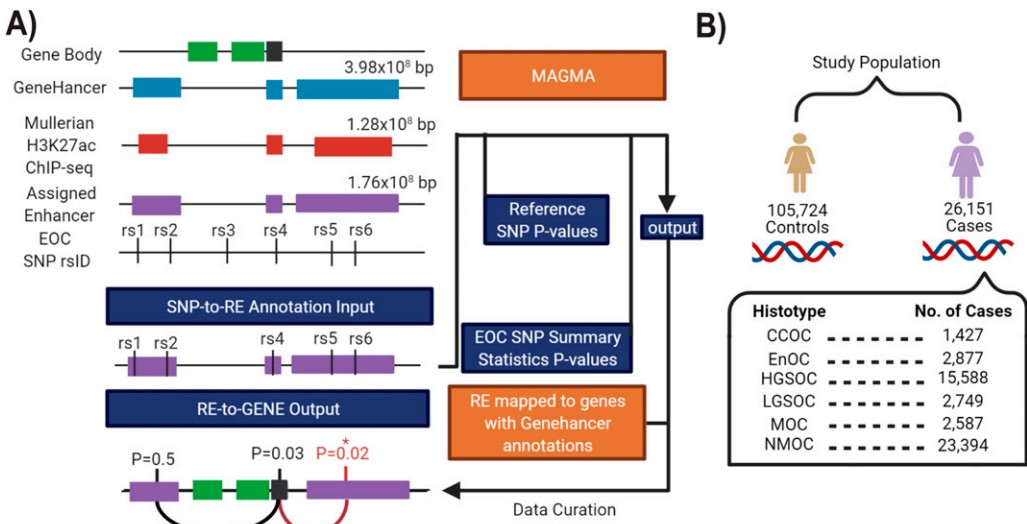

**Figure 1.  Applying chromMAGMA to epithelial ovarian cancer (EOC) risk.**
**(A)** Overview of the chromMAGMA approach. The GeneHancer database of regulatory elements (and linked genes) was limited to REs detected in Müllerian tissues. An EOC single-nucleotide polymorphism rsID-to-RE-to-gene annotation list was created and used for gene-level analysis using the MAGMA model, along with EOC genome-wide association study summary statistic $P$-values and reference linkage disequilibrium correlations from the European ancestry subset of the 1000 Genomes reference panel. Because multiple REs can be associated with one gene, the RE with the most significant $P$-value represents each gene. **(B)** Study population of EOC genome-wide association study dataset from Coetzee et al (2021) (Coetzee S, Dareng EO, Peng P, Rosenow W, Tyrer JP (2021) Integrative multi-omics analyses to identify the genetic and functional mechanisms underlying ovarian cancer risk regions. (Submitted for Publication) CCOC, clear cell ovarian cancer; EnOC, endometrioid ovarian cancer; HGSOC, high-grade serous ovarian cancer; MOC, mucinous ovarian cancer; NMOC, all non-mucinous ovarian cancers.

$\rho$ = −0.04, Fig S1). However, there was a high correlation in the ranked $P$-values before and after adjustment (Spearman's $\rho$ = 0.77) that was even higher in risk genes that passed the significant threshold (Spearman's $\rho$ = 0.98). Considering the aforementioned results in combination with the fact that chromMAGMA accounts for the number of SNPs mapped to each RE, this adjustment was excluded from the pipeline to achieve a balance between type II and type I errors.

We stratified REs into promoters (defined as 1,000-bp upstream and 100 bp downstream of a transcription start site) or candidate enhancers (all other regions of active chromatin), as both are marked by H3K27ac (see the Materials and Methods section). Of the 9,682 risk-associated REs identified for each histotype (range: 9,624–9,713), 38% of the REs (3,703/9,682; range: 3,669–3,731) were active promoters and 62% (5,979/9,682; range: 5,966–6,002) were enhancers. The enhancer-to-promoter distance varied widely, with an average distance of 187,647 bp (range: 2 bp–5 Mbp; SD ± 230,272 bp) between enhancer start and transcription start sites (Fig S2).

We ran conventional MAGMA alongside chromMAGMA to compare risk genes implicated by the two methods (Table S2). In MAGMA (and in chromMAGMA, given that the latter leverages the MAGMA statistical pipeline), the $P$-value is calculated in a two-step process: first the SNP matrix is projected into a smaller set of principal components to remove the effects of highly correlated SNPs; second these principal components are used in a linear regression whose outputs (feature-wise enrichment for significant SNPs) are tested for statistical significance using an F-test (6). Comparing the distributions of the population of $P$-values between MAGMA and chromMAGMA by bootstrapping revealed that chromMAGMA $P$-values are significantly lower than MAGMA across EOC histotypes (CCOC $P$-value = 9.6 × $10^{-5}$; EnOC $P$-value = 8.4 × $10^{-6}$; HGSOC $P$-value = 1.06 × $10^{-5}$; LGSOC $P$-value = 1.13 × $10^{-5}$; MOC $P$-value = 5.14 × $10^{-5}$; NMOC $P$-value = 8.00 × $10^{-6}$). This is consistent with previous evidence that REs, but not protein-coding exons, are enriched for risk-associated SNPs (3). After Bonferroni correction to account for the total number of genes tested in each histotype-specific analysis, we identified 68 unique significant genes in MAGMA and 155 unique significant genes in chromMAGMA, with 56 genes identified by both methods (MAGMA Bonferroni-corrected $P$-value < 2.70 × $10^{-6}$; chromMAGMA Bonferroni corrected $P$-value < 2.87 × $10^{-6}$). The number of genes identified by histotype ranged from 0 (EnOC) to 53 (all non-mucinous cancers, NMOC) significant genes in MAGMA and 0 (EnOC) to 131 (NMOC) significant genes using chromMAGMA (Fig 2A). Disparity in the number of significant genes by histotype is likely due to power, as HGSOC and NMOC represents a majority of the overall sample size in the EOC GWAS. To survey the functional role of chromMAGMA nominated risk genes, we leveraged a publicly available CRISPR-Cas9 knock-out screen that includes seven CCOC cell lines, four EnOC cell lines, 15 HGSOC cell lines, and five MOC cell line models (32). Dependency data were available for 149 out of 155 chromMAGMA nominated risk genes as dependency data for six genes were not available. This revealed 37 (out of 149) genes that are strong dependencies in 1 or more histological subtypes of EOC (Figs 2B and C and S3). Whereas most (29 out of 37) of these genes were considered essential in all histological subtypes of EOC represented, *HNF1B*, *SH3PXD2A*, *MEAF6*, and *SKAP1*

were dependencies specific to CCOC, EnOC, HGSOC, and MOC, respectively (Fig 2C and Table S3).

Gene-dense GWAS loci at genome-wide significance account for many of the risk genes identified by MAGMA and this is also the case for chromMAGMA (33). For example, genes on chromosome 17 are particularly overrepresented in MAGMA and chromMAGMA in EOC (26/68 and 52/155 unique genes, respectively) likely because of the presence of two genome-wide significant loci in this chromosome and the high degree of LD due to an inversion at 17q31 (13). We divided the genome into distinct bins based on LD to identify instances where chromMAGMA nominates candidate risk REs within distinct LD bins to the local risk association, scenarios where the same candidate gene could not readily be identified through MAGMA. Using this approach, 29 unique genes were identified as candidate risk genes only in chromMAGMA (Table S4). Using chromMAGMA NMOC as an example, a significant promoter ($P$-value 4.3 × $10^{-8}$) at a known breast and ovarian cancer genome-wide significant risk locus at chromosome 9q31 is assigned to *SMC2*, whereas in MAGMA, *SMC2* is not significant ($P$-value 1.9 × $10^{-3}$) (21) (Fig 2D). Other candidates not previously implicated in EOC risk such as *PRSS23* ($P$-value 2.6 × $10^{-6}$) was nominated by chromMAGMA, a serine protease regulated by HGSOC biomarker PAX8 (34), were also identified (Fig 2E). The same RE assigned to *PRSS23* interacts across an LD boundary with the promoter of *EED*. EED is a component of the polycomb repressor complex involved in the pathogenesis of numerous cancer types (35). In MAGMA, *PRSS23* and *EED* are not significant risk genes ($P$ = 0.44).

We next compared chromMAGMA risk genes with candidate susceptibility genes nominated by alternative approaches. This analysis was limited to HGSOC as it is the most common and well-studied EOC subtype. Chromosome conformation capture assays have identified candidate susceptibility genes previously undiscovered based on proximity to the nearest gene promoter. So far three GWAS significant loci at 11q31, 8q24, and 19p13 originally mapped by proximity to *HOXD3*, *PVT1*, and *BABAM1* was found, via chromosome conformation capture assays to interact with *HOXD9*, *MYC*, and *ABHD8*, respectively (14, 15, 36). chromMAGMA nominated all three as candidate susceptibility genes in HGSOC (*HOXD9*, $P$-value 1.52 × $10^{-12}$; *MYC*, $P$-value 6.39 × $10^{-11}$ and *ABHD8*, $P$-value 3.9 × $10^{-16}$). As chromMAGMA also identifies risk variants that may impact short-range enhancer–promoter interactions, promoters, and intronic enhancers, we reasoned that it should also be able to capture susceptibility genes nominated based on closest proximity to index SNP at risk loci. Indeed, 10 out of 12 (83%) genes previously identified as candidate risk genes based on proximity to a lead variant overlapped with chromMAGMA nominated genes ($P$-value < 5 × $10^{-8}$) (Fig 2F). *Cis*-expression quantitative trait loci (eQTL) and transcriptome-wide association studies (TWASs) leverage associations between risk SNP genotype data and gene expression to identify candidate genes associated with disease risk. To date, 26 candidate genes have been identified as HGSOC candidate risk genes using these methods (15, 37, 38); 16 out of 26 genes (62%) previously identified by HGSOC eQTL or TWAS analyses were also nominated by chromMAGMA (Fig 2F). chromMAGMA identified 105 additional genes previously not implicated in HGSOC risk. 22/105 of these genes (21%) had long-range interactions (>500 kb) with the risk RE, highlighting how chromMAGMA can identify candidate

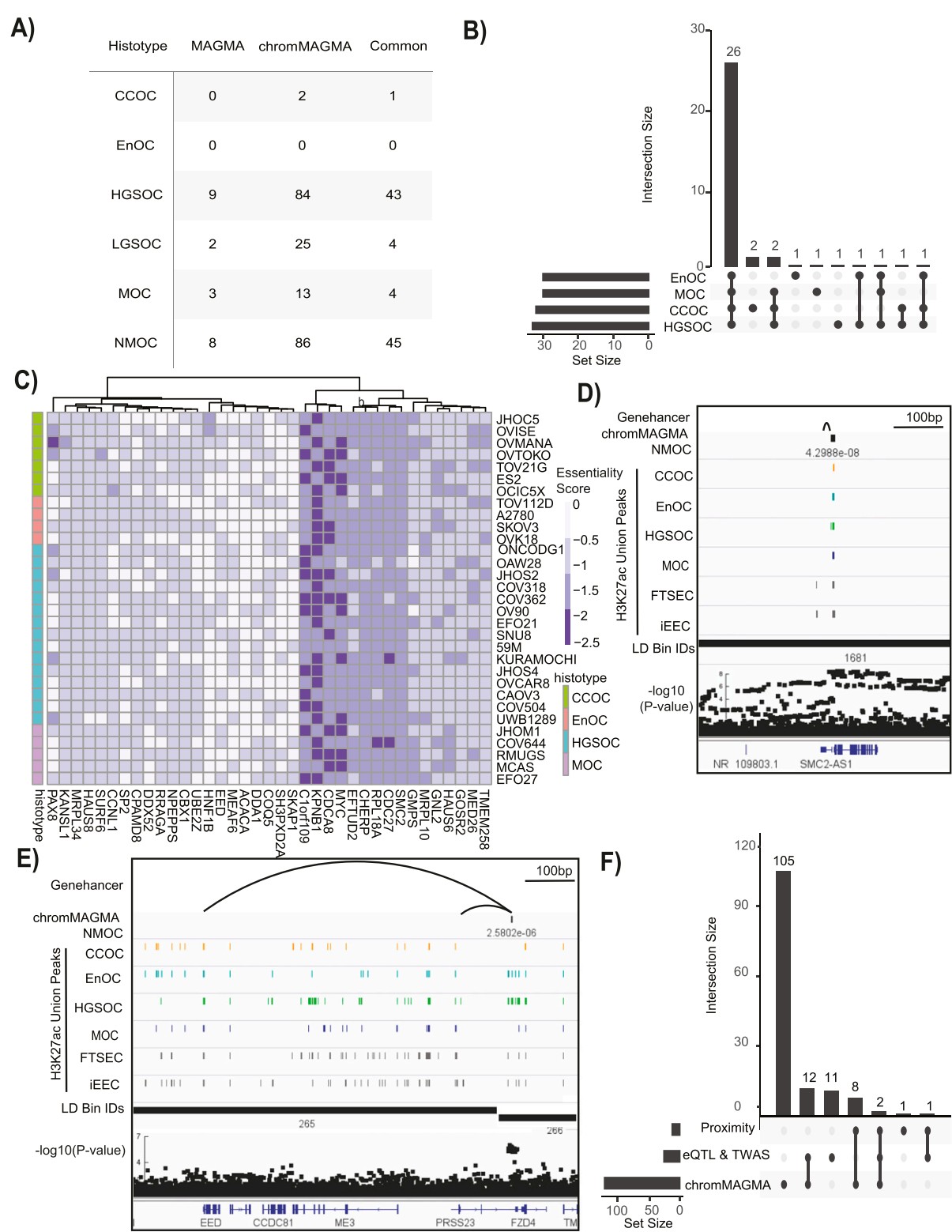

**Figure 2. ChromMAGMA identifies risk genes for epithelial ovarian cancer (EOC) through risk single-nucleotide polymorphisms coinciding with regulatory elements.**
**(A)** Candidate risk genes identified by conventional MAGMA and chromMAGMA across EOC histotypes (MAGMA Bonferroni corrected $P$-value < 2.70 × $10^{-6}$; chromMAGMA Bonferroni corrected $P$-value < 2.87 × $10^{-6}$). **(B)** Set analysis displaying 37 genes with strong dependency scores in one or more histological subtype of EOC. **(C)** Heat map displaying the dependency score of 37 chromMAGMA nominated genes with strong dependency scores. **(D)** Locus view displaying the NMOC chromMAGMA RE-to-gene association for *SMC2*. LD boundaries and genome-wide association study single-nucleotide polymorphism associations (−log₁₀[$P$-value]) are shown. **(E)** Locus view displaying the NMOC chromMAGMA RE-to-gene association for *PRSS23* and *EED*. **(F)** UpSet plot of chromMAGMA genome-wide significant genes in HGSOC and alternate

genes transcriptionally impacted by noncoding risk SNPs that are hundreds of kilobases away. For example, the longest promoter-risk RE interaction identified in this analysis was between the *GMPS* gene (*P*-value $5 \times 10^{-10}$) and an associated RE ~1 megabase away in linear genomic distance. Overall, chromMAGMA nominates candidate risk genes that are consistent with alternate methods but also implicates additional genes in HGSOC susceptibility through risk SNPs with their upstream active regulatory elements.

### chromMAGMA implicates splicing and gene regulation in EOC risk

To identify pathways regulated by risk-associated regulatory elements, we conducted gene set enrichment analysis to ask whether any gene sets from the Gene Ontology database were enriched in ranked gene-level associations (based on descending order of $-\log_{10}[P\text{-value}]$) from MAGMA and chromMAGMA. This approach allows for the investigation of sets of genes without the need to assign arbitrary *P*-value cutoffs. Because one RE can be assigned to multiple genes in the chromMAGMA gene-level association, gene ranks were weighted using an mRNA expression dataset comprising disease-relevant primary tissue samples (27) to generate a ranked list in which highly expressed genes are ranked higher than relatively lower expressed genes associated with the same RE (see the Materials and Methods section). Pathway enrichment analysis with the chromMAGMA-derived gene list identified 140 common pathways across all histotypes, of which seven were related to mRNA splicing and processing (considering only pathways with positive normalized enrichment scores and adjusted *P*-value < 0.05) (Fig 3A and Table S5). Spliceosome factors *CHERP* and *EFTUD2* were the top 2 (out of 349) most significant genes related to mRNA splicing in the weighted chromMAGMA gene list. In addition, 20 of the common pathways were terms related to transcription or chromatin, including RNA polymerase II activity and transcription factor activities (Fig 3A). DNA-binding transcription factors *SIN3B* and *NFE2L1* were the top 2 most significant genes (out of 1,746) in the weighted chromMAGMA gene list and the transcription mediator complex coactivator *MED26* ranked third. Histotype-specific pathways were also observed for CCOC (195 pathways), EnOC (58 pathways), HGSOC (17 pathways), LGSOC (30 pathways), and MOC (70 pathways) (Fig 3B). In contrast, pathway gene set enrichment with conventional MAGMA had no enriched pathways that passed the *P* < 0.05 (after adjustment for multiple comparisons) threshold across all histotypes.

### Super-enhancer–associated transcription factors are associated with EOC risk

TFs bind in a sequence-specific manner at promoters and enhancers to regulate gene expression, and both expression of TFs and TF binding sites are often involved in risk (12). Cancer cells are often dependent on TFs whose expression is propelled by large clusters of enhancers termed super-enhancers or stretch enhancers (39). Of 1,671 known human TFs, 257, 220, 202, and 247 TFs are associated with super-enhancers in CCOC, EnOC, HGSOC, and MOC, respectively (Table S6). Using chromMAGMA we identified super-enhancer–associated TFs as enriched for association with risk at *P*-value < 0.05; FDR (q-value) < 0.25 for all histotypes except LGSOC, as LGSOC tissue H3K27ac ChIP-seq data were not available (Fig 4A) (27, 40, 41). By contrast, super-enhancer–associated TFs were only significantly enriched for HGSOC risk (*P*-value < 0.05; FDR q-value < 0.25) when using gene-level statistics derived from conventional MAGMA. Leading-edge analysis was performed to identify the super-enhancer–associated TFs overrepresented in the top ranks of chromMAGMA gene-level associations. TFs previously implicated in EOC development including 6/14 candidate master regulators for HGSOC based on a recent pan-cancer gene expression analysis were implicated in EOC risk (Table S7). Three of these factors (PAX8, SOX17, and MECOM) are functionally validated master regulators of HGSOC development (Fig 4B) (42). *HNF1B*, a CCOC biomarker and a key regulator of CCOC tumorigenesis (43, 44), was also on the leading edge of the clear cell ovarian cancer analysis (Fig 4B).

### Gene set enrichment analysis of TF cistromes

In addition to TFs being the target of risk SNPs, noncoding SNPs may also impact disease risk by modifying TF binding within enhancers to impact gene expression (46, 47, 48). Therefore, we asked whether target genes of specific TFs are disproportionally impacted by EOC risk SNPs in chromMAGMA. For this analysis we asked if TF-specific gene sets in the Molecular Signatures database (MsigDB) are enriched in the ranked gene list from chromMAGMA (49, 50) (Table S8). TF targets are defined as genes with motifs located within 4 kb around their transcription start sites by MsigDB.

We first explored the PAX8 target gene sets in HGSOC, where we have previously identified an enrichment of PAX8 target gene sets in this histotype (51). PAX8 is represented by two gene sets in MsigDB: PAX8_B contains 106 genes and PAX8_01 contains 39 genes, with 23 genes in common across the two sets. When ranked by the normalized enrichment score, PAX8_B ranked 24/573 (*P*-value = 0.032, FDR q-value = 0.098) and PAX8_01 ranked 42/573 (*P*-value = 0.13, FDR q-value = 0.34) in MAGMA. With chromMAGMA, the PAX8 target gene sets ranked higher, with PAX8_B ranked 1/573 (*P*-value < $1.0 \times 10^{-3}$, FDR q-value = 0.115), and PAX8_01 ranked 12/573 (*P*-value = 0.040, FDR q-value = 0.18). We also explored chromMAGMA performed for NMOC, and this analysis identified targets of EVI1, also known as MDS1 and EVI1 Complex Locus (MECOM) as a significant gene set not identified in MAGMA (MsigDB-EVI1_05 *P*-value = $1.0 \times 10^{-3}$, FDR q-value = 0.17; MsigDB-EVI1_04; *P*-value = .041, FDR q-value = 0.15; MsigDB-EVI1_03; *P*-value = 4.6 × 10-2, FDR q-value = 0.229). MECOM is a known master regulator TF in HGSOC that is functionally involved in disease pathogenesis (52, 53). Leading-edge analysis was performed for MsigDB-PAX8_B and MsigDB-EVI1_05 to identify the candidate

approaches to nominate candidate risk genes. Proximity, lead variants labeled as genome-wide significant (P < $5 \times 10^{-8}$) assigned to genes based on nearest transcription start site; eQTL, *cis*-expression quantitative trait loci; TWAS, transcriptome-wide association studies; CCOC, clear cell ovarian cancer; EnOC, endometrioid ovarian cancer; HGSOC, high-grade serous ovarian cancer; LGSOC, low-grade serous ovarian cancer; MOC, mucinous ovarian cancer; NMOC, all non-mucinous ovarian cancers.

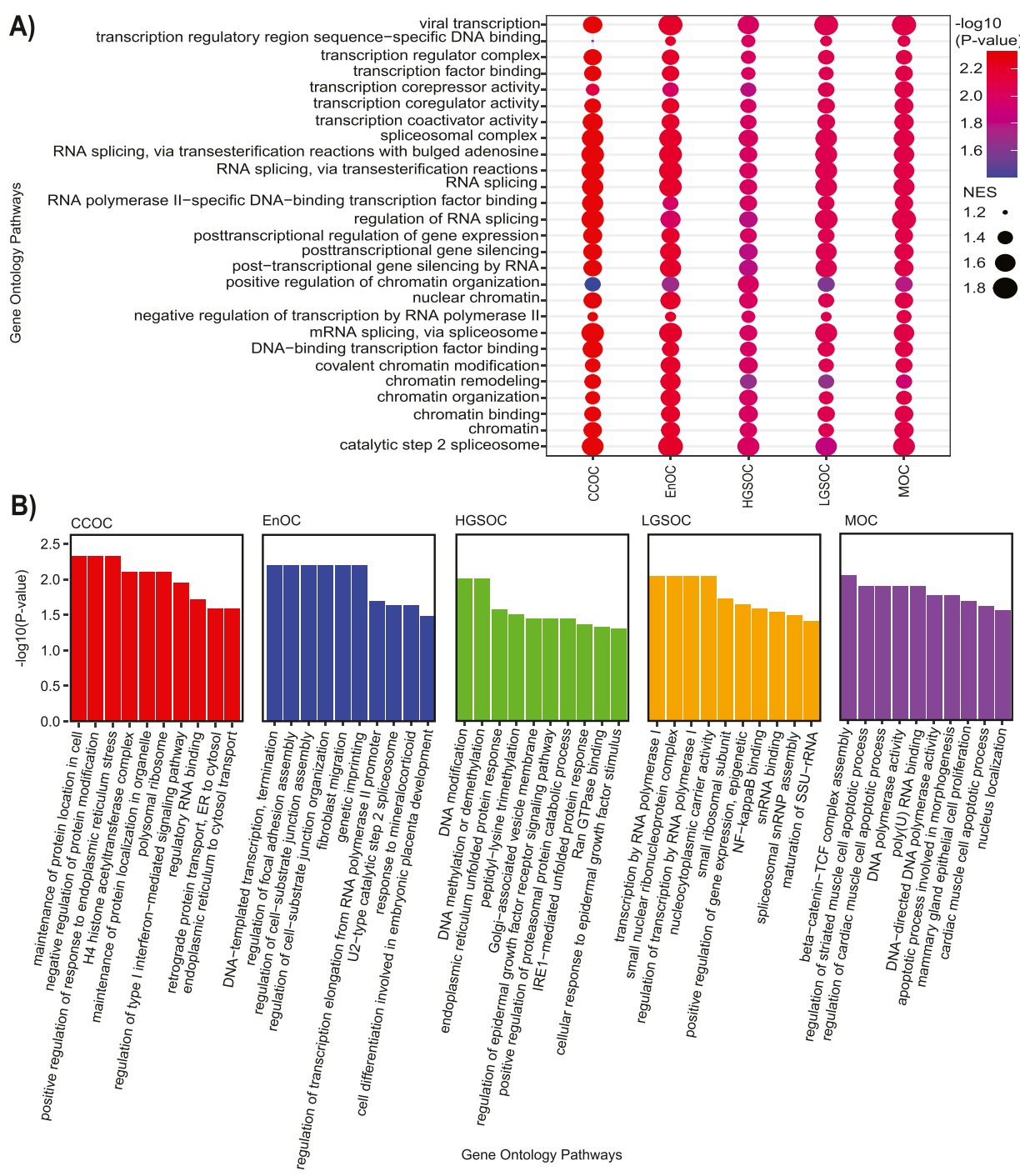

**Figure 3. chromMAGMA identifies histotype-specific as well as common pathways involved in epithelial ovarian cancer risk.**
**(A)** Dot plot representing transcription, splicing, and chromatin related pathways that were enriched in risk genes nominated in all histotypes by chromMAGMA. **(B)** Bar plots representing the top 10 histotype-specific chromMAGMA pathways based on normalized enrichment score. NES, normalized enrichment score; CCOC, clear cell ovarian cancer; EnOC, endometrioid ovarian cancer; HGSOC, high-grade serous ovarian cancer; LGSOC, low-grade serous ovarian cancer; MOC, mucinous ovarian cancer.

susceptibility genes potentially regulated by PAX8 and MECOM. This analysis identified 29 and 49 candidate susceptibility genes regulated by PAX8 and MECOM, respectively. *HOXB5*, *HOXB7*, *HOXB8*, and *NEUROD6* genes were common target genes between the two factors. *HOXB5*, *HOXB7*, and *HOXB8* are homeobox

superfamily TFs highly expressed in HGSOC and associated with poor survival (54).

We then used chromMAGMA to discover additional TFs not previously implicated in EOC risk with histotype specificity in consideration. Because one transcription factor can be represented

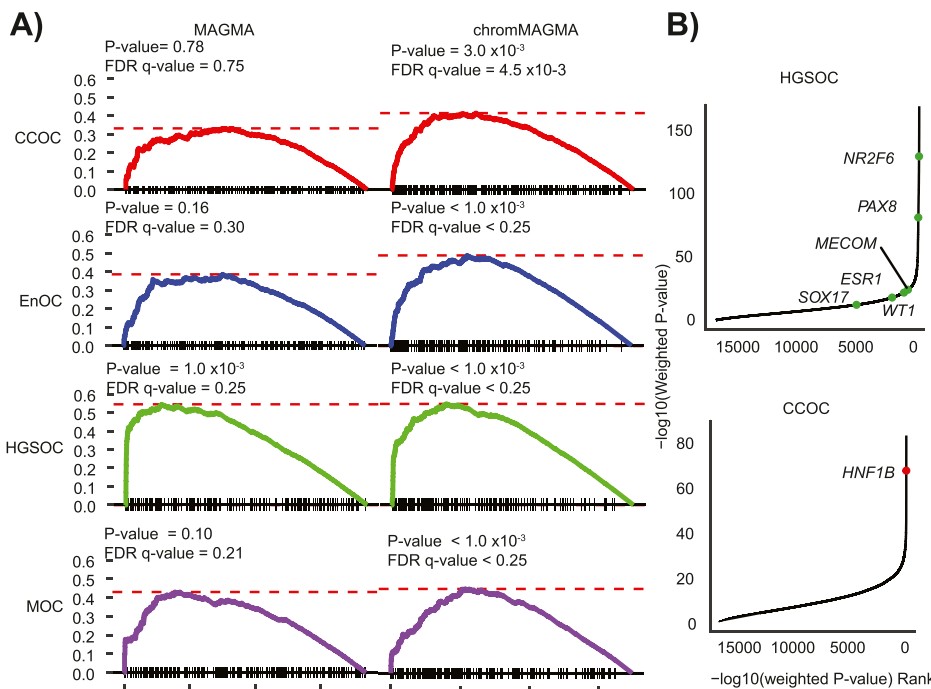

**Figure 4. Super-enhancers upstream of transcription factors are associated with histotype-specific epithelial ovarian cancer risk.** **(A)** Gene set enrichment plot of super-enhancer–associated TFs from each epithelial ovarian cancer histotype from Corona et al (2020) (45) in conventional MAGMA and chromMAGMA. **(B)** Gene –log₁₀(*P*-value) versus gene rank based on –log10(*P*-value) with known genes implicated in CCOC and HGSOC from the leading-edge list of the super-enhancer–associated TF gene set enrichment analysis highlighted. CCOC, clear cell ovarian cancer; EnOC, endometrioid ovarian cancer; HGSOC, high-grade serous ovarian cancer; LGSOC, low-grade serous ovarian cancer; MOC, mucinous ovarian cancer; NMOC, all non-mucinous ovarian cancers.

by multiple gene sets, the gene set with the most significant *P*-value was chosen to represent each transcription factor and, furthermore, NMOC was excluded from this analysis as this histotype dataset is a combination of all other subgroups except for MOC. Considering *P*-value <0.05 and FDR cutoff of < 0.25, we identified 113 transcription factors implicated in EOC risk. Of these 113 transcription factors, 13 were specific to CCOC, 4 to EnOC, 5 to HGSOC, 7 to LGSOC, 9 to MOC, and seven common across all five histotypes (Figs 5A and S4). *SOX9* was identified as a CCOC-specific TF in which its downstream regulatory targets are enriched for risk SNPs. A recent single-cell RNA sequencing study of the human endometrium (hypothesized tissue-of-origin for CCOC) grouped SOX9-positive epithelial cells of the endometrium as a regenerating and proliferative subset (55).

Finally, we set to identify TFs that are likely to be directly regulated by risk SNPs and where risk SNPs also modify TF downstream binding, hereinafter termed as "nexus TFs." Nexus TFs were defined as TFs that were (1) on the leading edge of the super-enhancer–associated TF gene set enrichment analysis and (2) TF target gene sets from MsigDB that were significantly enriched in chromMAGMA for each respective histotype (Fig 5B and Table S9). 16 TFs such as PAX8 were identified for HGSOC and EnOC, along with novel TFs implicated in EOC such as SP1, a TF implicated in a variety of biological processes across multiple cancer types (56). CRISPR-Cas9 knock-out screen from DepMap revealed that although there is heterogeneity across cell lines and histotypes, EOC lines are largely dependent on *RREB1*, *ATF4*, *MAX*, *PAX8*, *MZF1*, and *SRF* (average essentiality scores ≤ –0.4; average score for pan-essential genes = –1) and *SP1*, *MECOM*, and *CEBPB* (average essentiality scores ≤ –0.3) (Fig 5C and Table 1) (32). By comparison, negative control TFs that were (1) not on the leading edge of the super-enhancer–associated TF gene set enrichment analysis and (2) bottom 16 of the TF target

gene sets from MsigDB were less likely to be essential in EOC cell lines (Fig 5C). In total 9/16 nexus TFs showed at least modest dependency (average essentiality scores ≤ –0.3) in at least one histotype, compared with 2/16 negative control TFs. These results imply that TFs on the nexus of risk through genetic variation both in upstream REs and downstream binding sites can be identified in chromMAGMA and are often essential genes in EOC.

## Discussion

Most common risk polymorphisms associated with complex traits in GWAS are located in the noncoding portion of the genome (57). These noncoding risk polymorphisms likely modify the activity of noncoding regulatory elements to impact the expression of a target gene (or genes) that play a role in disease susceptibility (3). Identifying the risk RE and target gene remain two main challenges in post-GWAS functional work because REs outside of promoters are tissue specific and can interact with transcription start sites over large linear genomic distances (58). Here we built chromatin-MAGMA, or "chromMAGMA," to prioritize candidate risk REs and target genes based on the landscape of gene regulation in a specific tissue type. chromMAGMA first maps SNPs to user-defined tissue-specific regulatory element landscapes. REs are linked to likely target genes using the GeneHancer database, or any other resource. The collection of annotated active REs can then be interrogated at the gene level, or gene set level. Here we focused on epithelial datasets representing the tumor type of interest, plus likely precursor cell types; however, analogous datasets for other cell types could also be interrogated, where available.

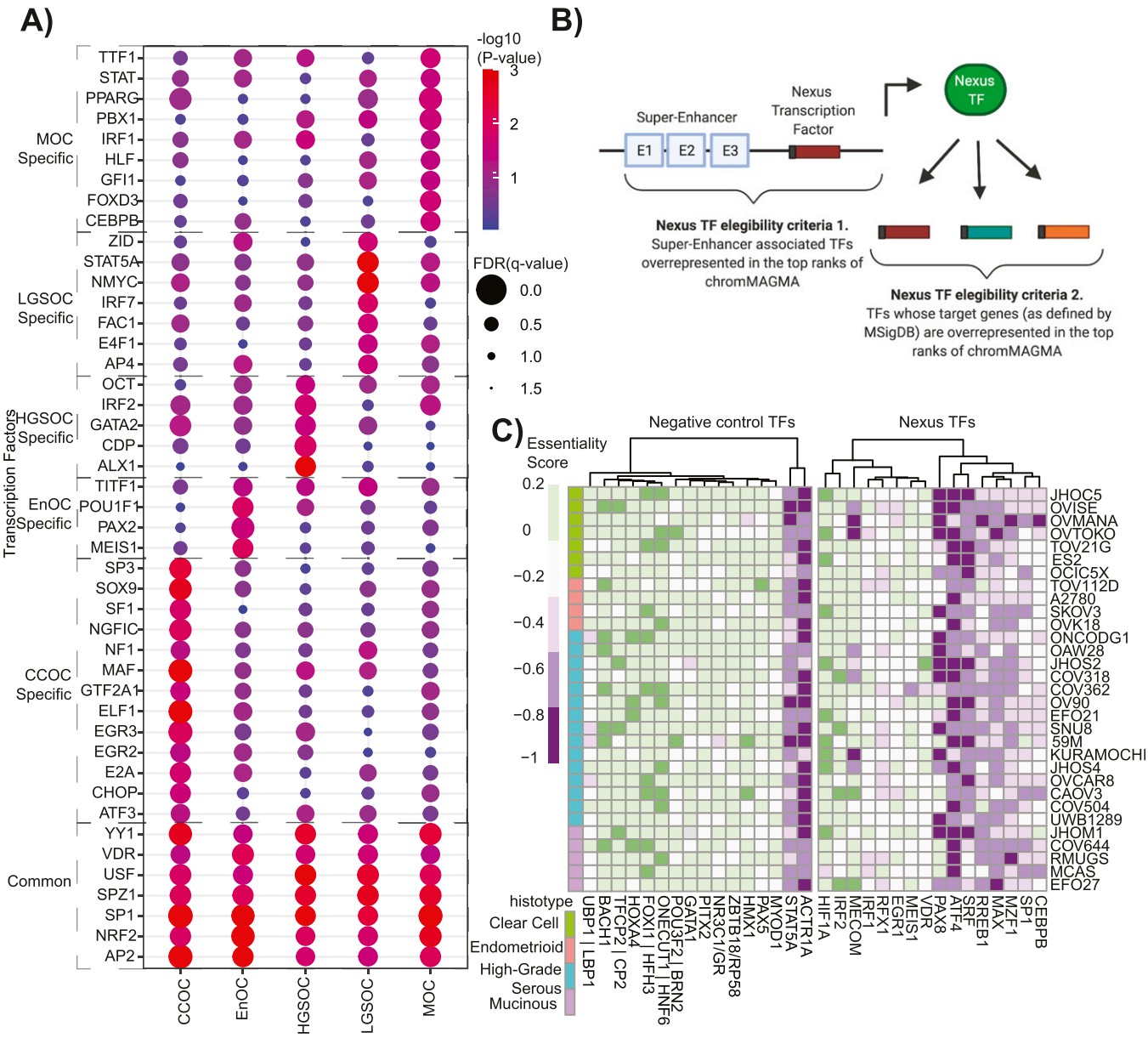

**Figure 5. Transcription factor networks in epithelial ovarian cancer risk.**
**(A)** Genome-wide significant Molecular Signatures Database transcription factor target (MsigDB "TFT_legacy") gene sets (*P*-value < 0.05 and FDR cutoff of <0.25) across EOC histotypes. **(B)** Schematic depiction of the definition of a "Nexus TF." Top ranks = leading edge of the gene set enrichment analysis. **(C)** Heat map displaying the essentiality score of Nexus TFs in epithelial ovarian cancer lines (data from Depmap.org). Columns clustered using unsupervised hierarchical clustering (method = K-means). CCOC, clear cell ovarian cancer; EnOC, endometrioid ovarian cancer; HGSOC, high-grade serous ovarian cancer; LGSOC, low-grade serous ovarian cancer; MOC, mucinous ovarian cancer; NMOC, all non-mucinous ovarian cancers.

We tested the performance of chromMAGMA using the largest EOC GWAS dataset to date, consisting of 26,151 EOC cases and 105,724 controls (Coetzee S, Dareng EO, Peng P, Rosenow W, Tyrer JP (2021) Integrative multi-omics analyses to identify the genetic and functional mechanisms underlying ovarian cancer risk regions. (Submitted for Publication) combined with disease-relevant Müllerian active REs and RE-to-gene contact maps from the Gene-Hancer database. We contrasted the RE-centric chromMAGMA to genes nominated by conventional MAGMA. Overall, chromMAGMA assigned lower *P*-values to genes compared to MAGMA, in

line with evidence that SNPs are enriched in active REs, validating the overall premise of this approach (2, 3, 4, 5). Orthogonal evidence to validate the chromMAGMA approach came from concordant results between chromMAGMA and alternative functional approaches to nominate candidate EOC susceptibility genes, including proximity, chromosome conformation capture assays, and quantitative trait locus-based analyses. Of particular note is that chromMAGMA identified previously validated candidate genes in scenarios where large genomic distances or multiple genes lie between the candidate causal risk SNPs and the risk gene. This highlights how the

**Table 1.  Average essentility scores for Nexus TFs across epithelial ovarian cancer cell lines.**

| TF | CCOC | | EnOC | | HGSOC | | MOC | |
|---|---|---|---|---|---|---|---|---|
| | Mean essentiality score | SD | Mean essentiality score | SD | Mean essentiality score | SD | Mean essentiality score | SD |
| EGR1 | −0.12 | 0.21 | −0.03 | 0.19 | −0.15 | 0.11 | −0.11 | 0.10 |
| RREB1 | −0.47[a] | 0.20 | −0.19 | 0.16 | −0.43[a] | 0.16 | −0.52[a] | 0.22 |
| SP1 | −0.34[b] | 0.22 | −0.37[b] | 0.23 | −0.29 | 0.20 | −0.50[a] | 0.08 |
| ATF4 | −0.76[a] | 0.18 | −0.73[a] | 0.06 | −0.78[a] | 0.20 | −0.91[a] | 0.26 |
| HIF1A | 0.34 | 0.36 | 0.15 | 0.10 | 0.14 | 0.10 | 0.06 | 0.30 |
| MECOM | −0.34[b] | 0.61 | 0.10 | 0.04 | −0.23 | 0.38 | 0.10 | 0.12 |
| MEIS1 | −0.01 | 0.06 | −0.04 | 0.16 | −0.11 | 0.16 | −0.04 | 0.15 |
| VDR | −0.05 | 0.13 | −0.09 | 0.14 | −0.06 | 0.12 | −0.13 | 0.07 |
| IRF2 | −0.04 | 0.09 | 0.04 | 0.04 | 0.10 | 0.18 | 0.02 | 0.21 |
| MAX | −0.58[a] | 0.16 | −0.47[a] | 0.10 | −0.52[a] | 0.15 | −0.62[a] | 0.12 |
| PAX8 | −0.98[a] | 0.66 | −0.36 | 0.40 | −0.71[a] | 0.58 | −0.25 | 0.44 |
| CEBPB | −0.40[b] | 0.25 | −0.18 | 0.05 | −0.30[b] | 0.13 | −0.28 | 0.32 |
| IRF1 | −0.16 | 0.10 | −0.17 | 0.13 | −0.09 | 0.07 | −0.20 | 0.13 |
| MZF1 | −0.56[a] | 0.28 | −0.48[a] | 0.17 | −0.51[a] | 0.11 | −0.58[a] | 0.23 |
| RFX1 | −0.15 | 0.11 | −0.25 | 0.14 | −0.26 | 0.11 | −0.24 | 0.18 |
| SRF | −0.76[a] | 0.17 | −0.52[a] | 0.16 | −0.67[a] | 0.14 | −0.48[a] | 0.29 |
| AR | −0.12 | 0.14 | −0.11 | 0.05 | −0.01 | 0.13 | −0.06 | 0.08 |

[a]Mean essentiality score ≤ −0.4.
[b]Mean essentiality score ≤ −0.3.
Mean essentiality score represents the average essentiality score for all cell lines associated with each epithelial ovarian cancer histotype. CCOC, clear cell ovarian cancer; EnOC, endometrioid ovarian cancer; HGSOC, high-grade serous ovarian cancer; LGSOC, low-grade serous ovarian cancer; MOC, mucinous ovarian cancer; NMOC, all non-mucinous ovarian cancers; SD, standard deviation.

chromMAGMA approach represents an efficient route to candidate gene nomination that incorporates the benefits of popular existing methods, while avoiding some of the limitations associated with those techniques. For example, chromMAGMA circumvents the distance bias of both eQTL analyses (which are often only powered to identify local *cis* interactions) or analyses that leverage chromatin interactome data (which conversely cannot resolve short-range interactions, which poses a particular challenge in gene-dense regions).

In addition to validating known risk genes, chromMAGMA provided insights into EOC risk that have not been achieved using previous methods. Pathway analysis of the chromMAGMA ranked gene list revealed enrichments of mRNA processing and splicing pathways across all histotypes, indicating that noncoding risk SNPs falling on REs regulate genes within these pathways. Whereas splicing events have been recently associated with EOC risk (59), components of splicing machinery have not been implicated in EOC risk previously. Transcriptional regulation pathways were also enriched in risk genes highly ranked by chromMAGMA, particularly super-enhancer–associated transcription factors (such as *PAX8* in HGSOC and *HNF1B* in CCOC). A study incorporating long-range, noncoding chromatin interactions from Hi-C with MAGMA (H-MAGMA) in nine neuropsychiatric disorders also found common pathways in transcriptional regulation/RNA splicing (60). These results suggest that risk variants impacting such pathways may be

common occurrences across complex traits. As TFs can be both targets and mediators of risk SNPs, we identified a set of "Nexus TFs," that is, transcription factors with oncogenic transcriptional properties that are enriched for risk variation both in its upstream *cis* regulatory element and in its downstream target binding sites. Gene dependency data prioritized nine transcription factors, which included master regulators of HGSOC–*PAX8* and *MECOM*. PAX8 and MECOM are known to co-occupy most of the chromatin marked by H3K27ac active regions in HGSOC and may be contributing to the differential regulation of HGSOC-relevant risk gene (42).

Whereas our study used H3K27ac chromatin immunoprecipitation data, a widely available mark of active chromatin, other technologies and epigenetic marks—such as other histone post-translational modifications, transcription factor binding sites, open chromatin regions, and methylation profiles—are all compatible with chromMAGMA. In this study, we used GeneHancer active regulatory-element-to-gene contact map data (28). GeneHancer is the most comprehensive catalogue of gene-regulatory element associations currently available and comprised RE-to-gene maps represented by 46 tissue types. One limitation to this approach is that Müllerian tissues are not well represented in the GeneHancer database and could be missing interactions unique to gynecologic tissues. Alternative data types, such as in silico maps of RE–promoter interactions inferred from ATAC-seq data (61) or genome-wide data from epigenome and genome editing screens could be

incorporated to create tissue-specific maps of gene-RE assignments. Another limitation of chromMAGMA is the necessary step of assigning a representative RE to a single gene for the generation of gene-level statistics. In this study, genes were mapped 1:1 to the RE with the most significant *P*-value. This step simplifies the biological complexity of multiple REs influencing a gene in an additive (62, 63), or sometimes hierarchal manner (64, 65), but for some genes, may miss a critical aspect of transcriptional regulation relevant to risk. Integration of chromMAGMA with data from perturbation (66), massively parallel reporter (67) assays and eRNA signatures (68) may be a superior way to prioritize REs associated with each gene. Overall, chromMAGMA is a flexible approach that can be readily adapted to prioritize candidate risk genes and regulatory elements for a wide array of phenotypes.

# Materials and Methods

### MAGMA

MAGMA uses the *P*-values of SNPs and local linkage disequilibrium to assign SNPs to gene locations, and then aggregates SNPs within the same gene body using a hypergeometric distribution (6). These genes are then ranked by ranking the $-\log_{10}(P\text{-value})$. The greater the $-\log_{10}(P\text{-value})$, the greater the number and/or significance of GWAS SNPs lying within the interval of the gene. MAGMA requires two external data sources: a list of GWAS SNPs with associated *P*-values from that of GWAS, number of participants, and a list of annotations linking gene names to intervals in the genome.

GWAS data came from the Ovarian Cancer Association Consortium (OCAC) study of 26,151 cases and 105,274 controls participants (Coetzee S, Dareng EO, Peng P, Rosenow W, Tyrer JP (2021) Integrative multi-omics analyses to identify the genetic and functional mechanisms underlying ovarian cancer risk regions. (Submitted for Publication). The GWAS data contained SNP *P*-values for five histotypes of ovarian cancer: high-grade serous, low-grade serous, clear cell, endometrioid, mucinous, and a composite category of all non-mucinous histotypes. Gene locations are from the NCBI build 37. Significant genes were identified by filtering genes whose *P*-values were less than the Bonferroni-corrected value of $2.70 \times 10^{-6}$.

### chromMAGMA

A list of all REs (hg19) and corresponding gene targets was obtained from GeneHancer (v4.7) a publicly available database of RE-to-gene maps (28). GeneHancer captures a broad universe of RE activity which we wished to reduce to those specific to ovarian cancer and precursor cell states. We used a dataset of H3K27ac peaks derived from clear cell (number of non-unique peaks = 119,549 peaks), endometrioid (125,743 peaks), high-grade serous (122,734 peaks), mucinous (131,655 peaks) ovarian tumor tissues, and samples from endometriosis epithelial (44,083 peaks), and normal fallopian tube secretory epithelial cell lines (43,734 peaks) (29, 45). This was converted to hg19 using UCSC liftOver, duplicates were removed and the remainder were merged into 80,271 distinct intervals using bedtools v2.25.0 (69). We then selected REs from GeneHancers

which overlapped with our H3K27ac intervals by at least one base pair.

We used the REs as the interval input into MAGMA to replace the gene intervals used by MAGMA. This generated a list of REs and their statistics. This list was then linked to the genes, where each gene was assigned the greatest $-\log_{10}(P\text{-value})$ from its REs. REs were defined as promoters based on the txdb.hsapiens.ucsc.hg19.knowngene database, and all non-promoters were labeled as candidate transcriptionally active enhancers. Significant genes were identified by filtering genes whose *P*-values were less than the Bonferroni corrected value of $2.87 \times 10^{-6}$.

### Identifying proximal genes to GWAS genome-wide significant loci

All lead variants labeled as genome-wide significant ($P < 5 \times 10^{-8}$) in ovarian cancer by Jones et al (2017) (13) were assigned to a gene based on nearest transcription start site.

### Generation of the gene list

Gene identifiers in chromMAGMA and MAGMA were curated by restricting to those identifiable as "ensembl_gene_id," "external_gene_name," "external_synonym," "hgnc_symbol," "entrez_gene_id," and "uniprot_gn_symbol" and filtered for genes labeled as "protein_coding" from the BioMart portal (70). For MAGMA, the maximal $-\log_{10}(P\text{-value})$ was then assigned to a gene, and simply ranked with $-\log_{10}(P\text{-value})$ in descending order. For chromMAGMA, ties in the $-\log_{10}(P\text{-value})$ were broken using the average expression of variance stabilization normalized primary CCOC, EnOC, HGSOC, MOC, and fallopian tube secretory epithelium (average Müllerian mRNA expression) as described from Corona et al. (2020) (45). The ties were broken using this formula:

$$\text{Weighted } P\text{-value} = -\log 10(P\text{-value})$$
$$\times \text{ Average Müllerian mRNA Expression.}$$

The same list was used for subsequent gene set enrichment analysis.

### Pathway gene set enrichment analysis

Pathway enrichment analysis was conducted using the ClusterProfiler package in R. We removed the HLA genes defined by the HLA Informatics Group (71, 72, 73) from the ranked list before carrying out gene set enrichment analysis. This is because the strong, long-distance linkage disequilibrium between SNPs in this region led to a clustering of multiple gene-level associations in this region making it difficult to differentiate between these genes in terms of ranks. This clustering in turn may yield potentially spurious enrichment signals for pathways that contain several HLA genes. We ran this analysis using the following script:

```
gseGO(geneList= <GENE-LIST>,
ont = "ALL",
keyType = "ENTREZID",
nPerm = 10,000,
minGSSize = 3,
```

```
        maxGSSize = 800,
        pvalueCutoff = 0.05,
        verbose = TRUE,
        OrgDb = org.Hs.eg.db,
        pAdjustMethod = "BH")
```

Pathways with gene set sizes less than 25 (as recommended by the BROAD institute) were removed from further analysis as the normalization to variation in gene set size becomes inaccurate for small gene sets.

### Super-enhancer–associated TF gene set enrichment analysis

This analysis was conducted using the default GSEA preranked setting within the Broad GSEA (v3.0) program. The super-enhancer–associated TF gene set was generated by taking known TFs from the Human Transcription Factor database (74), which was then filtered to only include TFs that were proximal, overlapping, or nearest to a super-enhancer as defined by ROSE2 (41) for CCOC, EnOC, HGSOC, MOC, and NMOC (45). Enrichment plots were generated with the R package fgsea.

### MsigDB TF cistrome gene set enrichment analysis

This analysis was conducted using the default GSEA-preranked setting within the Broad GSEA program (v3.0).

### Experimental methods

#### *Fallopian tube secretory epithelium RNA-seq*
RNA sequencing data from primary fallopian tube secretory epithelial cells were generated as described in Corona et al (2020) (45). They are available in the GEO database under the accession code GSE182510.

# Data Availability Statement

RNA-seq data can be found under accession number GSE182510.

### Code availability

A step-by-step tutorial of chromMAGMA is available in a Github repository: https://github.com/lawrenson-lab/chromMAGMA-public.

# Supplementary Information

# Acknowledgements

This project was supported by an Ovarian Cancer Research Fund Alliance Liz Tilberis Early Career Award (599175) (K Lawrenson), Ovarian Cancer Research Fund Alliance Program Project Development (373356) (K Lawrenson), a Southern California Clinical and Translational Science Institute Core Voucher (V148) (K Lawrenson). S Kar is supported by a United Kingdom Research and Innovation Future Leaders Fellowship (MR/T043202/1). The research described was supported in part by National Institute of Health (NIH)/National Center for Advancing Translational Science (NCATS) University of California, Los Angeles (UCLA) Clinical and Translational Science Institute (CTSI) Grant Number UL1TR001881. R Nameki is supported in part by a Ruth L Kirschstein Institutional National Research Service Award (T32) from the NIH (grant number 5 T32 GM 118288-2). Funding details for the Ovarian Cancer Association Consortium (OCAC) dataset used in this manuscript are to be found in Coetzee et al (2021) (Coetzee S, Dareng EO, Peng P, Rosenow W, Tyrer JP (2021) Integrative multi-omics analyses to identify the genetic and functional mechanisms underlying ovarian cancer risk regions. (Submitted for Publication), "Integrative multi-omics analyses to identify the genetic and functional mechanisms underlying ovarian cancer risk regions." NHS/NHS II was supported by NIH grants UM1 CA186107, P01 CA87969, R01 CA49449, U01 CA176726, and R01 CA67262. The content is solely the responsibility of the authors and does not necessarily represent the official views of the National Institutes of Health. NHS/NHS II thank the following state cancer registries for their help: AL, AZ, AR, CA, CO, CT, DE, FL, GA, ID, IL, IN, IA, KY, LA, ME, MD, MA, MI, NE, NH, NJ, NY, NC, ND, OH, OK, OR, PA, RI, SC, TN, TX, VA, WA, and WY. The authors take full responsibility for analyses and interpretation of these data. The NHS/NHS II study protocol was approved by the institutional review boards of the Brigham and Women's Hospital and Harvard TH Chan School of Public Health, and those of participating registries as required. The NHS/ NHS II acknowledge the Channing Division of Network Medicine, Department of Medicine, Brigham and Women's Hospital and Harvard Medical School, Boston, MA, USA, as the home of the Nurses' Health Studies. We also thank the following individuals and institutions from the Ovarian Cancer Association Consortium: Simon G Coetzee, Pei-Chen Peng, Will Rosenow, Stephanie Chen, Brian D Davis, Felipe Segato Dezem, Ji-Heui Seo, Alberto L Reyes, Katja KH Aben, Natalia N Antonenkova, Gerasimos Aravantinos, Laura E Beane Freeman, Matthias W Beckmann, Alicia Beeghly-Fadiel, Marcus Q Bernardini, Line Bjorge, Amanda Black, Natalia V Bogdanova, Kelly L Bolton, Agnieszka Budzilowska, Ralf Butzow, Hui Cai, Rikki Cannioto, Kexin Chen, AOCS Group, Yoke-Eng Chiew, Linda S Cook, Anna DeFazio, Joe Dennis, Jennifer A Doherty, Thilo Dörk, Andreas du Bois, Diana M Eccles, Gabrielle Ene, Peter A Fasching, James M Flanagan, Florentia Fostira, Aleksandra Gentry-Maharaj, Marc T Goodman, Christopher A Haiman, Florian Heitz, Michelle AT Hildebrandt, Estrid Høgdall, Claus K Høgdall, Ruea-Yea Huang, Michael E Jones, Daehee Kang, Beth Y Karlan, Anthony N Karnezis, Linda E Kelemen, Catherine J Kennedy, Elza K Khusnutdinova, Lambertus A Kiemeney, Susanne K Kjaer, Jolanta Kupryjanczyk, Diether Lambrechts, Melissa C Larson, Nhu D Le, Jenny Lester, Lian Li, Jan Lubiński, Michael Lush, Keitaro Matsuo, Taymaa May, John R McLaughlin, Francesmary Modugno, Melissa Moffitt, Steven A Narod, Tu Nguyen-Dumont, Håkan Olsson, N Charlotte Onland-Moret, Sue K Park, Jennifer B Permuth, Darya Prokofyeva, Harvey A Risch, Cristina Rodríguez-Antona, V Wendy Setiawan, Kang Shan, Melissa C Southey, Anthony J Swerdlow, Soo Hwang Teo, Kathryn L Terry, Pamela J Thompson, Liv Cecilie Vestrheim Thomsen, Cecilie F Torkildsen, Linda Titus, Britton Trabert, Ruth Travis, Shelley S Tworoger, Els Van Nieuwenhuysen, Digna Velez Edwards, Robert A Vierkant, Rayna Matsuno Weise, Nicolas Wentzensen, Stacey J Winham, Yin-Ling Woo, Li Yan, Wei Zheng, Argyrios Ziogas, Andrew Berchuck, Ellen L Goode, Celeste L Pearce, Susan J Ramus, Thomas A Sellers, Matthew L Freedman, Joellen M Schildkraut, Simon A Gayther, Dennis Hazelett, Michelle R Jones, and Jasmine T Plummer, MCCS, WMH. We also thank Pak Hin Yu for his technical assistance. Gene-essentiality results are generated by DepMap, BROAD https://doi.org/ 10.6084/m9.figshare.12280541.v4. Graphical abstract was generated with Biorender.com.

### Author Contributions

R Nameki: conceptualization, data curation, software, formal analysis, validation, investigation, visualization, methodology, project administration, and writing—original draft, review, and editing.
A Shetty: conceptualization, data curation, software, formal analysis, validation, investigation, visualization, methodology, and writing—original draft, review, and editing.

E Dareng: resources, data curation, and formal analysis.

J Tyrer: resources, data curation, and formal analysis.

X Lin: resources, data curation, and validation.

OCAC: resources, data curation, formal analysis, and funding acquisition.

P Pharoah: resources, data curation, formal analysis, and funding acquisition.

RI Corona: conceptualization, data curation, formal analysis, investigation, and methodology.

S Kar: conceptualization, supervision, funding acquisition, project administration, and writing—original draft, review, and editing.

K Lawrenson: conceptualization, supervision, funding acquisition, project administration, and writing—original draft, review, and editing.

## Conflict of Interest Statement

R Nameki, A Shetty, K Lawrenson, and S Kar report no conflicts of interest. Please refer to Coetzee et al (2021) (Coetzee S, Dareng EO, Peng P, Rosenow W, Tyrer JP (2021) Integrative multi-omics analyses to identify the genetic and functional mechanisms underlying ovarian cancer risk regions. (Submitted for Publication) for any conflicts of interests for members of the Ovarian Cancer Association Consortium.

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
