## [Reviewer comments · Life Science Alliance]

chromMAGMA: regulatory element-centric interrogation of risk variants

Robbin Nameki, Anamay Shetty, Eileen Dareng, Jonathan Tyrer, Xianzhi Lin, the Ovarian Cancer Association Consortium, Paul Pharoah, Rosario I. Corona, Siddhartha Kar, and Kate Lawrenson

DOI: 10.26508/lsa.202201446

Corresponding author(s): Dr. Kate Lawrenson (Cedars-Sinai Medical Center)

Review timeline:

Submission Date:	2022-03-11
Editorial Decision:	2022-03-14
Revision Received:	2022-05-19
Editorial Decision:	2022-05-24
Revision Received:	2022-06-07
Accepted:	2022-06-08

Scientific Editor: Eric Sawey

Transaction Report:

Please note that the manuscript was previously reviewed at another journal and the reports were taken into account in the decision-making process at Life Science Alliance. Since the original reviews are not subject to Life Science Alliance's transparent review process policy, the reports and author response cannot be published.

Re: Life Science Alliance manuscript #LSA-2022-01446-T

Dr. Kate Lawrenson
Cedars-Sinai Medical Center
Women's Cancer Program
Davis Room D3091
Los Angeles 90048

Dear Dr. Lawrenson,

Thank you for submitting your manuscript entitled "chromMAGMA: regulatory element-centric interrogation of risk variants" to Life Science Alliance. We invite you to submit a revised manuscript addressing the following Reviewer comments:

- Address Reviewer 1's comments, excluding points #5 and 6.
- Address Reviewer 2's comments via Discussion.
- Address Reviewer 3's comments.

Thank you for this interesting contribution to Life Science Alliance. We are looking forward to receiving your revised manuscript.

Sincerely,

- A letter addressing the reviewers' comments point by point.
- An editable version of the final text (.DOC or .DOCX) is needed for copyediting (no

PDFs).

B. MANUSCRIPT ORGANIZATION AND FORMATTING:

RE: Life Science Alliance Manuscript #LSA-2022-01446-TR

Dr. Kate Lawrenson
Cedars-Sinai Medical Center
Women's Cancer Program
Davis Room D3091
Los Angeles 90048

Dear Dr. Lawrenson,

Thank you for submitting your revised manuscript entitled "chromMAGMA: regulatory element-centric interrogation of risk variants". We would be happy to publish your paper in Life Science Alliance pending final revisions necessary to meet our formatting guidelines.

- please add ORCID ID for corresponding author-you should have received instructions on how to do so
- please remove the panel A for each of your supplementary figures, figure legends, and figure callouts; since this is the only panel in the figure, it does not need to be designated with a label
- please add a callout for Table S9 to your main manuscript text
- instead of "Accession Codes", please label this section "Data Availability Statement" and repeat here that this accession is for the RNA-seq data

A. FINAL FILES:

B. MANUSCRIPT ORGANIZATION AND FORMATTING:

Sincerely,

RE: Life Science Alliance Manuscript #LSA-2022-01446-TRR

Dr. Kate Lawrenson
Cedars-Sinai Medical Center
Women's Cancer Program
Davis Room D3091
Los Angeles 90048

Dear Dr. Lawrenson,

Thank you for submitting your Research Article entitled "chromMAGMA: regulatory element-centric interrogation of risk variants". It is a pleasure to let you know that your manuscript is now accepted for publication in Life Science Alliance. Congratulations on this interesting work.

DISTRIBUTION OF MATERIALS:

Again, congratulations on a very nice paper. I hope you found the review process to be constructive and are pleased with how the manuscript was handled editorially. We look forward to future exciting submissions from your lab.

Sincerely,
